# General Nonlinearities in SO(2)-Equivariant CNNs

**Daniel Franzen**
Institute of Computer Science
Johannes Gutenberg University Mainz
Staudingerweg 9,
55122 Mainz, Germany
dfranz@uni-mainz.de

**Michael Wand**
Institute of Computer Science
Johannes Gutenberg University Mainz
Staudingerweg 9,
55122 Mainz, Germany
wandm@uni-mainz.de

## Abstract

Invariance under symmetry is an important problem in machine learning. Our paper looks specifically at equivariant neural networks where transformations of inputs yield homomorphic transformations of outputs. Here, steerable CNNs have emerged as the standard solution. An inherent problem of steerable representations is that general nonlinear layers break equivariance, thus restricting architectural choices. Our paper applies harmonic distortion analysis to illuminate the effect of nonlinearities on Fourier representations of SO(2). We develop a novel FFT-based algorithm for computing representations of non-linearly transformed activations while maintaining band-limitation. It yields exact equivariance for polynomial (approximations of) nonlinearities, as well as approximate solutions with tunable accuracy for general functions. We apply the approach to build a fully E(3)-equivariant network for sampled 3D surface data. In experiments with 2D and 3D data, we obtain results that compare favorably to the state-of-the-art in terms of accuracy while permitting continuous symmetry and exact equivariance.

## 1   Introduction

Modeling of symmetry in data, i.e., the invariance of properties under classes of transformations, is a cornerstone of machine learning: Invariance of statistical properties over samples is the basis of any form of generalization, and the prior knowledge of additional symmetries can be leveraged for performance gains. Aside from data efficiency prospects, some applications require exact symmetry. For example, in computational physics, symmetry of potentials and force fields is directly linked to conservation laws, and is therefore important for the stability of simulations.

In deep neural networks, (discrete) translational symmetry over space and/or time is exploited in many architectures and is the defining feature of convolutional neural networks (CNNs) and their successors. In most applications, we are typically interested in invariance (e.g., classification remains unchanged) or co-variance (e.g., predicted geometry is transformed along with the input). Formally, this goal is captured under the more general umbrella of equivariance [6]:

Let $f : X \to Y$ be a function (e.g., a network layer) that maps between vector spaces $X, Y$ (e.g., feature maps in a CNN). Let $G$ be a group and let (in slight abuse of notation) $g \circ v$ denote the application of the action of group element $g$ on a vector $v$. $f$ is called *equivariant*, iff:

$$\forall g \in G : f(g \circ v) = h(g) \circ f(v), \tag{1}$$

where $h : G \mapsto G'$ is a group homomorphism mapping into a suitable group $G'$. Informally speaking, the effect of a transformation on the input should have an effect on the output that has (at least) the same algebraic structure. Invariance ($h \equiv 1_{G'}$) and covariance ($h = id_{G \to G'}$) are special cases, along with contra-variance and any other isomorphisms of subgroups of $G$.

35th Conference on Neural Information Processing Systems (NeurIPS 2021).

**G-CNNs:** To make current CNN architectures (consisting of linear layers and nonlinearities) equivariant, the standard (and also most general, see [40]) approach are *group convolutional networks* (G-CNNs) [6], which conceptually boil down to just applying all transformations $g \in G$ to filters, correlating the result with the data, and storing the results. Typically, $G$ will be a continuous, compact Lie group such as SO($d$) (our paper focuses on SO(2)). To avoid infinite costs, results are band-limited, i.e., stored as coefficients of a truncated Fourier basis on $G$. Simultaneously, the Fourier coefficients provide a linear representation of a subgroup of $G$ and thus exact equivariance in the sense of Eq. 1. Using such a basis to directly construct sets of filters yields steerable filter banks [41], were each filter outputs a whole vector of coefficients that represent functions on $G$.

**Nonlinearities:** Unfortunately, band-limiting interferes in non-trivial ways with network architecture, as applying a standard nonlinearity such as $\mathrm{ReLU}, \tanh$, or even simple nonlinear polynomials to the Fourier coefficients directly will break equivariance. Multiple solutions to this problem have been proposed [39]: Multiplicative nonlinearities [40; 22] as in tensor networks keep equivariance but behave differently from traditional nonlinearities and therefore cannot be used as a drop-in in classical CNN architectures. Complex nonlinearities such as $\mathbb{C}-\mathrm{ReLU}$ that only act nonlinearly on the magnitude of Fourier coefficients [43; 42] also keep perfect equivariance but are less expressive as they do not permit nonlinear operations on the phase information. A recent study by Weiler and Cesa [39] shows that simple discretized rotations [6], which do not require architectural adaptations but provide only approximate equivariance, yield the best practical results in image classification tasks.

**SO(2)-equivariance:** The goal of our paper is to clarify the effect of nonlinearities on Fourier-domain representations. We restrict ourselves to the case of SO(2), which permits the application of standard harmonic distortion analysis [26]. Our goal is to maintain a band-limited representation of a function on SO(2) (corresponding to a fixed angular resolution) and efficiently compute the band-limited Fourier-representation after application of a nonlinearity. We obtain an exact algorithm for polynomial nonlinearities with a computational overhead of $\mathcal{O}(D \log D)$ for degree $D$. For general nonlinearities, we observe quick convergence in numerical experiments.

**SE(3)-equivariant surface networks:** While the limitation to SO(2) might appear restrictive, it is still important for many applications processing image and geometric data: Adding translational invariance (SE(2)-equivariance) is easy, and we also apply our representation to surface data with normal information, extending ideas of Wiersma et al. [42] to a fully E(3)-equivariant network.

We evaluate our networks on example benchmarks for 2D image and 3D object recognition. We obtain invariant results and equivariant intermediate representations up to numerical precision for polynomial nonlinearities (double checking the formal guarantees) and low-error approximations with general nonlinearities such as $\tanh, \mathrm{ReLU}, \mathrm{ELU}$ [5] at reasonable overhead (typically less than 20%, depending on approximation quality). Classification accuracy on MNIST-rot and ModelNet-40 [44] is roughly similar to other recent literature for $\mathrm{ReLU}$ and slightly reduced for low-degree polynomial approximations [16] of $\mathrm{ReLU}$.

Our main contributions are (i) a simple analytical model for the effect of nonlinearities on Fourier representations in SO(2)-equivariant networks and (ii) an efficient algorithm for applying nonlinearities. It is provably exact for polynomials and empirically yields good approximations for common non-polynomial functions. This permits, for the first time, a the usage of standard CNN architectures with common nonlinearities without compromising equivariance.

## 2   Related Work

**2D rotational equivariance:** The are various approaches to achieving equivariance or invariance to rotations and other transformations in CNNs. The first approaches focused on exact $C_4$-equivariance of rotated images [23] by applying the same network to rotated copies of the input (with shared weights), followed by some invariant operation (e.g. max-pooling), after which the feature maps become invariant to input rotations. More advanced architectures allow the network to "see" filter responses of the previous layer to different rotations of the inputs by adding weights for those inputs in a way that does not break equivariance [6; 7; 10]. More fine-grained equivariance can be achieved by representing the image on a hexagonal grid [19], or by using *steerable filters* [41; 4] or Fourier representations [43]. A comprehensive study that puts these different approaches into a common framework and systematically analyzes their performance is provided by Weiler and Cesa [39].

**Processing 3D data:** While early neural network models for 3D data operated on regular grids similar to their 2D counterparts, alternative methods using more sparse representations for processing 3D data with neural networks have quickly emerged. Some methods rely on data reduction through lower dimensional projections, such as generating rendered images (sometimes from multiple perspectives to improve performance and approximate equivariance) and feeding them into into classical 2D-CNNs [34]. Other methods achieve rotational equivariance by projections on spherical surfaces, on which convolution operations can be defined. These surfaces can be represented by a sampling (e.g. CSGNN[13] uses an icosahedral sampling) or a spherical harmonics basis [11].

Graph-based networks perform convolutions on a connectivity graph. For example, SchNet [32] is a popular architecture for predicting molecular properties, operating on the graph of molecular bonds. MeshCNN [17] defines convolution operations directly on the vertices of a 3D triangle mesh and uses vertex merging as pooling operation. Other methods work directly on the intrinsic geometry of manifolds [42]. Such methods often have the advantage of being naturally equivariant to rotations or translations, as they work on intrinsic properties of an object. However, they often need to be tailored to a specific type of data.

**Point-based methods:** As a more general approach, point-based architectures have become a popular alternative. Early models like PointNet [29] and follow-ups [30; 25; 38] directly used the coordinates of 3D data as network input. Later, Point Convolutional Neural Networks [1] emerged as an attempt to generalize the grid-based CNNs architecture to a spatial convolution on point clouds, using a radial Gaussian filter basis with fixed or trainable offset values. Kernel Point Convolutions (KPConv) [36] improve on this by introducing correlation functions which define the interaction of nearby points and limiting the distance of interactions to reduce overhead.

Many approaches to construct rotationally equivariant point-based architectures rely on specially designed convolution operations that produce rotationally invariant feature vectors, which can then be processed further without restrictions (e.g. any kind of nonlinearity can be used without breaking equivariance). One way to achieve this is aligning the inputs to the convolutional filters by using so-called local reference frames, as done GCANet [46], and by MA-KPConv [35], which extends the KPConv [36] model by using multiple alignments for the filters. Other approaches, like ClusterNet [3], LGR-Net [47], RI-Conv [45] and RI-Framework [24] work on rotation invariant local or global representations of the input point could. Invariant features can also be obtained by applying an *invariant map* to the output of an equivariant convolution operation, as in Spherical Harmonics Networks (SPHnet) [28], which calculate activations in a spherical harmonic basis and produce invariant output by taking the norm over coefficients with identical degree.

**Equivariant representations:** Various methods use equivariant representations throughout their architecture. For example, Tensor Field Networks [37] and 3D Steerable CNNs [40] generalize the notion of steerable representations to 3-dimensional data. Deng et al. [9] propose "*Vector Neurons*", a general framework for transforming non-equivariant point-based architectures like PointNet [29] or DGCNN [38] to equivariant ones by replacing scalar activations with 3D vectors. Such methods mostly use specially designed nonlinarities. In contrast, we propose an FFT-based algorithm that allows to apply general nonlinearities on equivariant representations and yields exact equivariance for polynomial functions. In recent concurrent work, de Haan et al. [8] also discuss FFT-based approximate equivariance in the context of gauge invariance on meshes (without considering polynomial functions specifically).

Our SE(3)-equivariant architecture is probably closest to the approach of Wiersma et al. [42], that differs from the methods above by giving each feature vector its own local reference frame, which is aligned to the surface normal and one arbitrarily chosen perperdicular direction. Equivariance is guaranteed by using parallel transport along the surface to align the reference frames of different features. In contrast to this, we skip the parallel transport step and use a simpler method that aligns the local coordinate systems by finding a common tangent vector, which only requires knowledge of the normal vectors associated with each point.

Our main analytical tool is harmonic distortion analysis. Originally developed in physics and engineering [12], Mehmeti-Göpel et al. [26] have recently applied it to the problem of understanding the trainability of deep networks. In our case, we study how our linear representations of SO(2) are affected by using a similar transformation with respect to an angular variable $\alpha$.

# 3 SO(2)-Equivariant Networks

We start with a very brief recap of SO(2)-equivariant steerable networks [7; 39]. We formulate the approach in terms of band-limited angular functions rather than steerable filter banks. This is merely a transposed view, but facilitates the discussion in Section 4.

Our starting point is a network layer $f^{(l)}, l \in \{1, ..., L\}$ that maps functions $x^{(l-1)}$ to functions $x^{(l)}$. These feature functions $x$ depend on a spatial location $\mathbf{t} \in \mathbb{R}^2$, and additionally, as SO(2)-equivariant CNNs index the channels by the continuous group SO(2) [4], an angle parameter $\alpha$ that represents these rotations in an arc-length parametrization:

$$x^{(l)} : \mathbb{R}^2 \times \mathbb{R} \mapsto \mathbb{R}^{C^{(l)}} \quad \text{with periodicity} \quad x^{(l)}(\mathbf{t}, \alpha) = x^{(l)}(\mathbf{t}, \alpha + 2\pi) \ \forall \, \mathbf{t} \in \mathbb{R}^2, \alpha \in \mathbb{R} \quad (2)$$

In general, $x^{(l)}$ has vector-valued output, representing multiple feature channels as in classical CNNs. In the following, we assume a single input and output channel ($C^{(l)} = 1$) for simplicity.

Analogous to discrete rotation equivariant networks [6; 10], where feature channels are associated with a cyclic rotation group and perform a cyclic shift when the input is rotated, we define an operation $T\gamma$ that periodically shifts our continuous SO(2) features concurrently with a rotation in the spatial domain. Using $\Theta_\gamma$ to denote the rotation of a 2D vector by an angle $\gamma$, this transformation can be written as:

$$T_\gamma \, x^{(l)}(\mathbf{t}, \alpha) = x^{(l)}(\Theta_\gamma \mathbf{t}, \alpha - \gamma) \quad (3)$$

According to Eq. 1, equivariance under this transformation is guaranteed if the network layers $f^{(l)}$ commute with $T_\gamma$. The inputs of our first layer are usually scalar feature maps without dependency on an angle $\alpha$. In this case, equivariance under $T_\gamma$ can be achieved by convolving our input $x^{(0)}(\mathbf{t})$ with all rotated copies of our trainable filter $w$:

$$\hat{x}^{(1)}(\mathbf{t}, \alpha) = \int_{\mathbb{R}^2} d\mathbf{t}' \, w^{(1)}(\Theta_\alpha(\mathbf{t}' - \mathbf{t})) \, x^{(0)}(\mathbf{t}') \quad (4)$$

After each layer, a nonlinearity $\varphi$ is applied to the *pre-activations* $\hat{x}^{(l)}$, using a trainable bias $b$, to generate the final layer output $x^{(l)}$. In the continuous angular representation, $\varphi$ is applied for each angle $\alpha$ independently and therefore does not interfere with equivariance as defined in Eq. 1:

$$x^{(l)}(\mathbf{t}, \alpha) = \varphi\left(\hat{x}^{(l)}(\mathbf{t}, \alpha) + b^{(l)}\right) \quad (5)$$

From the second layer on, our inputs are now also angle-dependent. While modifying Eq. 4 by adding the parameter $\alpha$ to $x$ on the right hand side would preserve equivariance, the resulting network architecture would perform independent calculations for each angle $\alpha$ and therefore exhibit no interaction between different filter rotations in consecutive layers.

To actually take advantage of the equivariant design, we need to allow the layers of our network to also "see" the filter responses for different rotation angles $\alpha$ of the previous layer. This can be achieved by adding an additional SO(2)-dependency to our weights $w$. However, simply making $w$ periodically dependent on $\alpha$ would not work—to preserve equivariance, we need to perform a joint convolution over the domains $\mathbb{R}^2$ and SO(2), as noted by Cheng et al. [4]:

$$\hat{x}^{(l)}(\mathbf{t}, \alpha) = \int_{\mathbb{R}^2} d\mathbf{t}' \int_0^{2\pi} d\alpha' \, w^{(l)}(\Theta_\alpha(\mathbf{t}' - \mathbf{t}), \alpha' - \alpha) \, x^{(l-1)}(\mathbf{t}', \alpha') \quad (6)$$

Equivariance of Eq. 6 under $T_\gamma$ can be verified by putting the transformation $T_\gamma$ before $x$ on both sides, and showing that the resulting equation is equivalent to the original one. The proof is provided in the supplementary material.

## 3.1 Fourier representation

To make our SO(2)-dependent layer activations $x$ representable in finite memory, we assume that they are band limited to a maximum frequency of $K^{(l)}$. According to the sampling theorem [14], such functions can be represented exactly by a uniform sampling with $2K^{(l)} + 1$ samples, e.g. by using a complex Fourier series:

$$x^{(l)}(\mathbf{t}, \alpha) = \sum_{k=-K}^{K} z_k^{(l)}(\mathbf{t}) \, e^{ik\alpha}, \quad z_k \in \mathbb{C} \quad (7)$$

To use a similar Fourier representation for our weights $w$, we first decompose the spatial dependency on $\mathbb{R}^2$ into a product of a radial profile $\rho$ and an angular function $q$, using $\angle(\mathbf{t})$ to denote the signed angle of a 2D vector $\mathbf{t}$ relative to the $x$-axis of the coordinate plane:

$$w^{(l)}(\mathbf{t}, \alpha) = \rho^{(l)}(\|\mathbf{t}\|) \cdot q^{(l)}(\alpha, \angle(\mathbf{t})) \tag{8}$$

The angular part $q$ can then, as it depends on two angles, be represented in a 2D Fourier series (for now, we use an infinite series here, however, only matching coefficients will be used later):

$$q^{(l)}(\alpha, \beta) \sum_{u,v \in \mathbb{Z}} q_{u,v}^{(l)} e^{iu\alpha} e^{iv\beta}, \quad q_{u,v} \in \mathbb{C} \tag{9}$$

Plugging the Fourier representations from Eq. 7, 8 and 9 in Eq. 6 allows us to calculate the convolution directly on the Fourier coefficients (the derivation is included in supplementary material):

$$\hat{z}_k^{(l)}(\mathbf{t}, \alpha) = \int_{\mathbb{R}^2} d\mathbf{t}' \, \rho^{(l)}(\|\mathbf{t}' - \mathbf{t}\|) \sum_{k'=-K}^{K} q_{-k',k-k'}^{(l)} \, z_{k'}^{(l-1)}(\mathbf{t}') \, e^{i(k-k')\angle(\mathbf{t}'-\mathbf{t})} \tag{10}$$

Note that as we use the Fourier basis to represent real-valued functions in the spatial domain, the conjugation symmetry holds for activations ($\overline{z_k} = z_{-k}$) as well as weights ($\overline{q_{u,v}} = q_{-u,-v}$). Thus, in our concrete implementation, we only store half of the coefficients. Angle-independent input (e.g. in the first layer) is passed to the network via the zero-order Fourier coefficient ($z_0^{(0)} = x^{(0)}$).

Although not immediately apparent, the individual complex Fourier coefficients $z_k$ in Eq. 7 are equivalent to the "irrep" representations of same order $k$ used by Weiler and Cesa [39]. However, different from their approach, we always look at multiple Fourier coefficients of different order as part of a series, describing a real-valued function defined on SO(2). Band-limiting, i.e., truncation of the Fourier series, maintains a linear representation of a subgroup of SO(2), yielding an equivariant network (otherwise, the construction so far would be fully covariant for invertible convolutions).

## 4   Nonlinearities in SO(2)-Equivariant Networks

In the continous case, general nonlinearites do not cause problems—equivariance still holds (trivially) if the nonlinearity $\varphi$ is applied point-wise, at every angle, as defined in Eq. 5. However, in the Fourier representation, the activation value for a specific angle $\alpha$ is now spread over multiple Fourier coefficients. Thus, applying general nonlinarities would not only require a transformation back from the frequency into the angular domain, but also a re-encoding, which involves a continuous integral that requires a potentially expensive numerical calculation: Unlike linear mappings, a nonlinear mapping can create higher-order frequencies, called "harmonics".

### 4.1   Nonlinearities create harmonics

In order to understand this effect, we apply harmonic distortion analysis [12; 26]. To simplify the notation, we will in the following drop the layer index and denote by $\hat{z} = [\hat{z}_{-K}, ..., \hat{z}_K]$ the series of Fourier coefficients of the pre-activation $\hat{x}(\alpha)$ of a single feature channel at a fixed layer. For the analysis, we assume that $\varphi$ is a polynomial of finite degree $D$:

$$\varphi(\hat{x}) = \sum_{j=0}^{D} c_j \hat{x}^j \tag{11}$$

Plugging in the corresponding Fourier series in the polynomial yields:

$$\varphi\left(\sum_{k=-K}^{K} \hat{z}_k e^{ik\alpha}\right) = \sum_{j=0}^{D} c_j \sum_{k_1,...,k_j=-K}^{K} \hat{z}_{k_1} \cdots \hat{z}_{k_j} e^{i(k_1+\cdots+k_j)\alpha} \tag{12}$$

The convolution theorem now converts the point-wise multiplication into a convolution of the spectra in the Fourier domain: The series $z \in \mathbb{C}^{\mathbb{Z}}$ of output Fourier coefficients is given by

$$z = c_0 + c_1 \hat{z} + c_2(\hat{z} * \hat{z}) + c_3(\hat{z} * \hat{z} * \hat{z}) + \cdots + c_D(\hat{z} * \cdots * \hat{z}) \tag{13}$$

where "$*$" denotes the discrete convolution

$$[\mathrm{z} * \mathrm{w}]_k := \sum_{m \in \mathbb{Z}} z_m \cdot w_{k-m}. \tag{14}$$

As expected, an application of a nonlinear function could potentially spread the spectrum towards higher frequencies. Inputs band-limited to frequency $K$ yield outputs band-limited to $KD$.

We can use this observation to construct efficient algorithms for computing z from $\hat{\mathrm{z}}$. Our goal is to compute the first $K'$ Fourier coefficients after applying the nonlinearity efficiently from the $K$ Fourier coefficients of the pre-activations. While the convolutional layers of our network correspond to a complex matrix-vector multiplication, applying the activation function is not trivial due to the presence of negative frequencies in the Fourier series, that lead to a mixing of high and low frequency components. Furthermore, a naive computation on only $K'$ coefficients could introduce aliasing effects that break equivariance.

## 4.2 Exact equivariance: Polynomial nonlinearities

**Direct convolution:** The simplest correct solution is to directly and iteratively evaluate the discrete convolution in Eq. 13 $D$-times, requiring $\mathcal{O}(\sum_{j=1}^{D} K \cdot jK) = \mathcal{O}(K^2 D^2)$ time and $\mathcal{O}(DK)$ temporary memory, with only $\mathcal{O}(K')$ values being kept in the end.

**FFT-based algorithm:** The direct algorithm becomes inefficient for higher-order polynomials. The alternative is to evaluate $\varphi(\hat{x})$ directly in the angluar domain, which requires an inverse Fast Fourier Transformation (IFFT), application of $\varphi$ and a forward FFT. For large $D$, the point-wise application of $\varphi$ in the angular domain is obviously more efficient than the Fourier-domain convolution. However, in order to maintain equivariance, we need to sample adequately.

Counting the non-zero coefficients in Eq. 14 shows us immediately that $2DK + 1$ Fourier coefficients, i.e., a Fourier expansion up to frequency $DK$, is sufficient for an exact evaluation of a degree $D$ polynomial: Arising from a $D$-fold convolution in the Fourier domain, the signal is band-limited accordingly. The sampling theorem then translates this to equidistant sampling at $2DK + 1$ discretization points in the angular domain (the corresponding discrete FFT becomes a bijection). This gives us an asymptotically more efficient algorithm with run-time $\mathcal{O}(DK \log DK)$ and memory $\mathcal{O}(DK)$ that is still exact. As the discrete Fourier transformation is a unitary map, we can also expect stable numerical properties. For low orders $D$, direct convolution might be slightly more efficient than the FFT-based approach. Nonetheless, our current implementation uses only the FFT variant for simplicity, and also because we observe that the overhead of applying the nonlinearities is minor in relation to the overall computational costs of our networks.

**Practical application:** As any polynomial of degree $D \geq 1$ diverges asymptotically, neural networks using polynomial nonlinearities can become unstable during the training process [16], resulting in exploding gradients and activations. This is especially problematic for higher-order polynomials. We address this by limiting the $\ell_1$-norm of the Fourier coefficients at a maximum value, keeping $\|\hat{z}\|_1 \leq \hat{x}_{\max}$ with $\hat{x}_{\max}$ chosen bounding the range the polynomial was designed for. The $\ell_1$-norm is an upper bound of the maximum value $x(\alpha)$ for all $\alpha$ that is also tight, as can be seen by an example Fourier series with aligned complex phases such that $|\hat{x}(0)| = \sum_{k=-K}^{K} |\hat{z}_k|$.

## 4.3 Approximation: Oversampled FFT

For non-polynomial nonlinearities, the FFT-based algorithm can still be used for approximate evaluation. We still map to the angular domain and back via FFT, using $D$-fold oversampling, but $D$ now becomes a user-parameter. In the general case, the spectrum will not be band-limited, but decaying (the Fourier series converges for functions of bounded variation, which certainly applies to post-activations in all practical networks). Thus, the truncated FFT will create (non-equivariant) aliasing artifacts that should vanish with increasing $D$.

The convergence behavior depends on the fall-off of the Fourier spectrum for a nonlinearity that is not bounded in polynomial degree, which is hard to quantify. Even if we assume a polynomial approximation, we would need large degrees, and results would depend on the decay rate of the coefficients $c_j$ employed to gain realistically tight bounds. Empirically, we do observe a quick increase of precision with oversampling, as shown in Section 6. As the run-time costs of the FFT are moderate, this algorithm is useful in applications requiring only approximate equivariance.

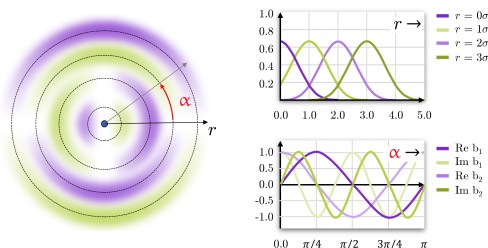

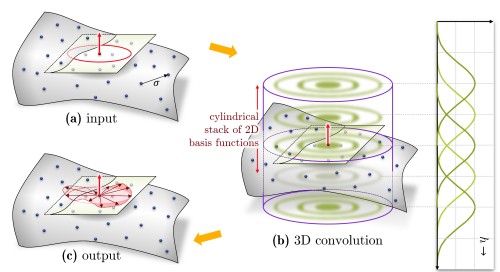

Figure 1: *SE(2)-equivariant point networks:* Each point is associated with a set of concentric SO(2)-steerable filters (Fourier basis), modulated by equidistant gaussians in radial direction (for translational band-limiting).

Figure 2: *SE(3)-equivariant surfel networks*: (a) for oriented surfels, we (b) perform 3D convolutions with unknown rotation around the normal direction, outputing (c) a collection of functions defined on tangential directions.

## 5 Application to Geometric Data

In the following, we apply the SO(2)-equivariant network layer described in Section 3 to 2D and 3D data, using the FFT-based approach defined in Section 4 for applying various nonlinearities.

### 5.1 SE(2)-Equivariant Networks for 2D Point Clouds

We construct an SE(2)-equivariant network for a set of input points $\mathbf{p}_1^{(0)}, ..., \mathbf{p}_N^{(0)}$, where each point carries an arbitrary input feature vector $\mathbf{x}_n^{(0)}$. The points can be a regular pixel grid of an image or a sampling of a geometric object. We equip each point $\mathbf{p}_n$ with a set of radial filters in polar coordinates $(r, \phi)$, such that a 2D filter corresponds to a set of concentric circles of radius $r > 0$, on each of which a trainable Fourier basis in $\phi$ is used to linearly represent SO(2) (Figure 1). Radially, we discretize by fixed FIR low-pass filters—our current implementation uses equidistant Gaussians

$$\rho_d(r) = \exp\left(\frac{(r - r_0 d)^2}{2\sigma^2}\right), \quad d \in \{1, ..., D\}. \tag{15}$$

Figure 1 illustrates this construction. The input points for each layer are treated as Dirac functions. The exact formulation of our point-based architecture is provided in the supplementary material.

**Further layer types:** For constructing complete networks, we employ various custom-built layers. To gather activations in the spatial domain, we employ average pooling, which averages Fourier coefficients (which is possible due to linearity of the Fourier transform). As *invariant map* [39], we either output only the scalar Fourier coefficient $z_0$ in the last convolutional layer (*conv2triv*) or take the norm of all complex outputs of the last convolutional layer after the nonlinearity has been applied. Batch normalization [20] of Fourier representations follows the obvious route of obtaining the mean via the $z_0$-coefficients and the variances via the power spectrum $||\mathbf{z}||_2^2$ of the Fourier coefficients. Instead of tracking a running mean during training for batch normalization, we calculate the exact training set statistics after training in one extra pass, while not changing other weights.

**Nonlinearities:** We evaluate our network for various general nonlinearities (ReLU, $\tanh$ and ELU [5]) and polynomial approximations of ReLU of degrees 2 and 4 (see Figure 4) taken from Gottemukkula [16], computed by the FFT algorithm outlined in section 4. In case of the polynomials, we clamp the $\ell_1$-norm of each channel's Fourier coefficients to the range $[-5, 5]$ before applying the nonlinearity to avoid problems with exploding activations or gradients. We also include the $\mathbb{C}$–ReLU function [43] in our experiments (which acts on the norm of the activations only and therefore requires no Fourier transformation) to better estimate the performance penalties of the FFT method.

**Architecture:** Our concrete network design follows the construction of Weiler and Cesa [39] for their best MNIST-rot models. We use the same number of equivariant and linear layers with the same output channel count and filter properties (radii, rotation orders and width) and also apply Dropout [33] with $p = 0.3$ before each linear layer. We train our models for 40 epochs with a batch size of 64 images. We use the Adam [21] optimizer, starting with a learning rate of 0.015, with an exponential decay factor of 0.8 per epoch starting after epoch 16. We calculate the mean test error and its standard deviation from 10 independent training runs with He-initialized [18] weights.

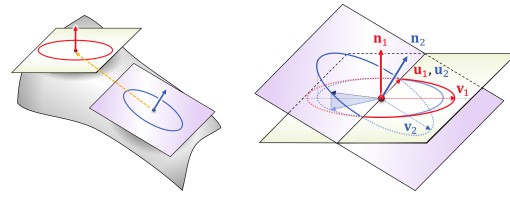

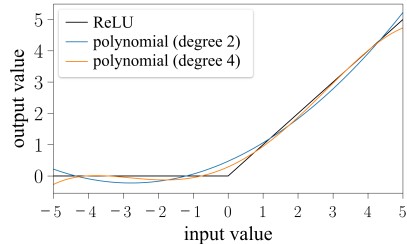

Figure 3: Vector features are projected into the tangent space of the target point.

Figure 4: Polynomial approximations of the ReLU function [16].

Table 1: Results on the MNIST-rot dataset

| model | group | repre-sentation | num. coeff. | FFT pad | activation function | invariant map | model param. | sec / epoch | test error (%) mean | std |
|-------|-------|-----------------|-------------|---------|---------------------|---------------|--------------|-------------|------|-----|
| E2CNN [39] | $C_{16}$ | regular | 16 | - | ELU | *maxpool* | 2,692,690 | 38 | 0.716 | 0.028 |
| E2CNN [39] | $C_{16}$ | quotient | 16 | - | ELU | *maxpool* | 2,749,686 | 49 | 0.705 | 0.045 |
| E2CNN [39] | $D_{16\|5}C_{16}$ | regular | 16 | - | ELU | *maxpool* | 3,069,353 | 76 | 0.682 | 0.022 |
| Ours | SO(2) | Fourier | 9 | - | $\mathbb{C}$–ReLU | *norm* | 1,396,138 | 30 | 0.980 | 0.031 |
| Ours | SO(2) | Fourier | 9 | 127 | ELU | *norm* | 1,394,986 | 36 | 0.729 | 0.029 |
| Ours | SO(2) | Fourier | 9 | 127 | tanh | *norm* | 1,394,986 | 36 | 0.768 | 0.024 |
| Ours | SO(2) | Fourier | 9 | 127 | ReLU | *norm* | 1,394,986 | 36 | 0.685 | 0.026 |
| Ours | SO(2) | Fourier | 9 | 7 | ReLU | *norm* | 1,394,986 | 30 | 0.689 | 0.019 |
| Ours | SO(2) | Fourier | 17 | 127 | ReLU | *norm* | 2,729,098 | 64 | 0.699 | 0.033 |
| Ours | SO(2) | Fourier | 9 | 127 | ReLU | *conv2triv* | 891,178 | 36 | 0.719 | 0.018 |
| Ours | SO(2) | Fourier | 9 | 8 | Poly(2)–ReLU | *norm* | 1,394,986 | 32 | 0.690 | 0.015 |
| Ours | SO(2) | Fourier | 9 | 24 | Poly(4)–ReLU | *norm* | 1,394,986 | 36 | 0.690 | 0.024 |

## 5.2 SE(3)-Equivariant Surfel Networks

Following the concept of Wiersma et al. [42], we can employ SO(2)-equivariant layers for building SE(3)-equivariant surface networks. Our construction differs slightly from theirs: It uses extrinsic rather than intrinsic operations, thus actually creating exact SE(3)-equivariance but not having isometric ("bending") invariance. We consider point-sampled surfaces with oriented normals at every point ("surfels", [27]) and perform the same construction as in the 2D point cloud case, equipping each individual point with vertical stack of radial filters aligned with its tangent plane (see Figure 2a,b). In vertical direction, we use the same Gaussian filters as in radial direction (for consistent band-limiting). The filters are convolved against the point clouds, with uniform features $\mathbf{x}_n = (1)$ used as input for all $n$ in the first layer. The resulting angular functions can be interpreted as functions defined on tangential directions (i.e., on unit-vectors in the tangent plane).

When performing a second level of convolution, we need to relate the tangential feature functions computed by different surfels, which live in different reference frames. We achieve this by aligning both angular functions to a common reference frame, defined by a common tangent vector. This allows us to apply a projection to our complex coefficients, which is explained in detail in the supplementary material. This construction is purely geometric and thus covariant under rigid transformations (the whole geometry is just rotated/translated together). Specifically for coefficients of rotation order 1, which can be viewed as tangential vectors fixed to the object, this approach can be illustrated as projecting these vectors to the tangential plane of the output point (see Figure 3).

## 6 Results

We implemented SO(2)-equivariant layers and the corresponding SE(2)- and SE(3)-equivariant point cloud / surfel networks in PyTorch using PyKeOps [2] for computing the sparse matrix-vector multiplications in general point clouds efficiently. We discuss the most important aspects regarding our experiments in this section. A more comprehensive presentation, along with source code, is provided in the supplementary material.

**Data sets:** We test our implementation on image and 3D data. In the image case, we replicate the architecture used by Weiler and Cesa [39] on the MNIST-rot dataset from their recent survey of

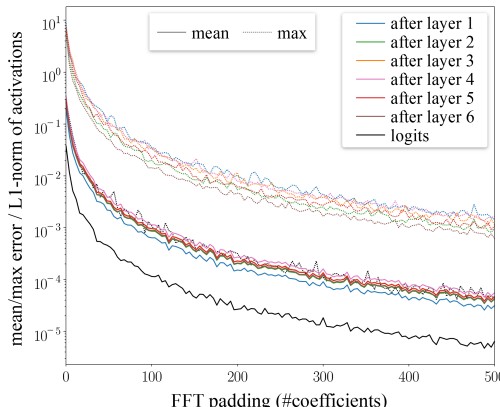 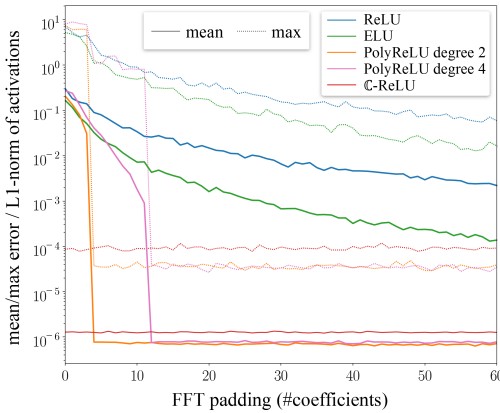

Figure 5: Relative error of ReLU activations (basic MNIST-rot network, 9 Fourier coeffs., *norm*-map) for random rotations vs. unrotated input. solid: mean absolute error, dashed: maximum error, relative to the layer-wise L1-norm for batches of 32 images.

Figure 6: Same errors as in Figure 5 after the fifth (penultimate) equivariant layer for various nonlinearities. Polynomials show the expected sharp decline at with increasing FFT padding. $\mathbb{C}$–ReLU included as reference (no FFT used). Note the different scale on the axes.

Table 2: Results on the ModelNet-40 dataset

| method | model training time | test accuracy (%) | | |
| --- | --- | --- | --- | --- |
| | | N/SO(3) | $z$/SO(3) | SO(3)/SO(3) |
| Spherical CNNs [11] | n.a. | - | - | 81.3 |
| RI-Conv [45] | 3 hrs (GTX 1080) | - | 86.4 | 86.4 |
| ClusterNet [3] | 9.5 hrs (unknown) | - | 87.1 | 87.1 |
| SPHnet [28] | 1.5 hrs (RTX 2080 Ti) | 86.6 | - | 87.6 |
| CSGNN [13] | 6.8 hrs (Tesla P100) | - | 86.2 | 88.9 |
| MA-KPConv [35] | 7.5 hrs (Tesla V100) | 89.1 | 89.1 | 89.1 |
| GCANet [46] | n.a. | - | 89.1 | 89.2 |
| RI-Framework [24] | n.a. | 89.3 | 89.4 | 89.3 |
| VN-DGCNN [9] | n.a. | - | 89.5 | 90.2 |
| SFCNN [31] | n.a. | - | 85.3 | 91.0 |
| LGR-Net [47] | n.a. | - | 90.9 | 91.1 |
| Ours (Poly(2), FFT pad 8) | 55 min (RTX 2080 Ti)* | 88.0 | 88.0 | 88.0 |
| Ours (Poly(4), FFT pad 24) | 60 min (RTX 2080 Ti)* | 88.7 | 88.7 | 88.7 |
| Ours (ReLU, FFT pad 127) | 60 min (RTX 2080 Ti)* | 89.1 | 89.1 | 89.1 |

*: times do not include dataset preprocessing and point cloud generation

**E(2)-equivariant image processing networks.** For simplicity, we convert all image data to point clouds. This comes with some computational overhead, but the absolute training times are still comparable to the pixel-based implementation used in [39], see Table 1. The point-based representation can be rotated exactly, facilitating the measurement of accuracy in terms of equivariance.

For the 3D surfel case, we use ModelNet-40 [44] as benchmark. We rescale all models to a unit bounding cube and convert the polygonal data to point clouds by z-Buffer rasterization from 50 random view points. Normals are estimated from a PCA-fit to 20 nearest neighbors at a sample spacing of 0.005, and oriented to point away from the center of mass. The final input points for our networks are obtained by a reduction using Poisson-disc sampling with a sample spacing of 0.05.

**Accuracy of Equivariance** Figure 5 shows the error on rotated inputs for the FFT-sampled ReLU using various amounts of oversampling (padding applied to the Fourier basis) for all layers of the MNIST-rot network. The equivariant layers show a similar relative error, with the error on the final invariant (logits) layer being lower. All experiments use 32-bit floating-point GPU computations.

We compare the equivariance error of different nonlinearities in Figure 6. In accordance with our theoretical considerations from Section 4.2, the error for the polynomial nonlinearities drops sharply when a specific amount of oversampling is applied. From this point on, further oversampling does not improve equivariance, which suggests that the remaining small fluctuations are due to the numerical limitations of the 32-bit floating point representation. This is supported by the observation that

$\mathbb{C}$–ReLU, which should be perfectly equivariant, as it only operates on the norm of the Fourier coefficients, produces a similar level of fluctuations.

The errors for the approximations of ReLU and ELU drop continuously with increasing oversampling, with ELU dropping significantly faster that ReLU. The convergence behavior is good enough to reach reasonable stability to input rotations (mean output errors of $10^{-5}$) with feasible oversampling (up to 512 coefficients). For high accuracy requirements, the polynomial approach appears to be favorable, with an order of magnitude lower error.

**Prediction Accuracy:** Table 1 lists our results on the MNIST-rot dataset, together with those obtained by Weiler and Cesa [39], which we managed to reproduce using their published code. Our models using ReLU or its polynomial approximations reach comparable performance to the best rotation-only ($C_{16}$) equivariant models evaluated this paper, while having a lower overall parameter count.

The results on ModelNet-40 are shown in Table 2. As common in the literature, we refer to a model trained on the original (non-augmented) dataset and evaluated with random rotations as N/SO(3), while we denote rotational augmentation during training and testing as SO(3)/SO(3). Additionally, we evaluated our model for $z$/SO(3), denoting random rotations around the $z$-axis during training and SO(3) rotations during testing, which is also used in some papers. The resulting accuracy of our surfel based model is roughly similar to other architectures when using the ReLU nonlinearity, while polynominal activation functions perform slightly worse.

**Approximate vs exact equivariance:** For the considered classification tasks, the approximate FFT seems provides stable enough equivariance, as a larger oversampling, while indeed reducing the numerical error (see Figure 6), has negligible impact on classification accuracy (see Table 1). This is also supported by the error levels for the final logits in Figure 5, which are an order of magnitude lower than for the intermediate equivariant layers. However, while we limit ourselves in this paper to common classification benchmarks to allow for comparisons with the literature, accurate equivariance is much more important in many other applications such as physics simulations, where symmetry (of predicted potentials or forces) corresponds to conservation laws [15].

**Computational Cost** In Table 1, we list training time in seconds per epoch on a single *Nvidia RTX 2080 Ti* graphics card. We can estimate the compuational cost of the FFT by comparing the runtimes with those of the $\mathbb{C}$–ReLU nonlinearity, which can be quickly computed without performing an FFT. This suggests that the overhead for the FFT is low (smaller than 20%, depending on the amount of oversampling) compared to the other operations performed. Note that the times per epoch also reflect total training time, as all models (E2CNN [39] and Ours) are trained for 40 epochs.

# 7    Conclusions & Future Work

In this paper, we have presented an analysis of the effect of nonlinear activation functions on the Fourier representations used by SO(2)-equivariant networks. The main insight is that the nonlinearity creates high frequency harmonics; thus applying the nonlinearity to an oversampled angular domain representation can maintain equivariance. For polynomial nonlinearities, this construction is provably exact. This theoretical prediction is also observed in a real numerical implementation. For general nonlinearities, we empirically observe quick convergence. As a sanity check, we have applied our method to shape and image classification, and reach comparable performance to other recent literature, while providing full, continuous equivariance.

Our main result, the oversampling algorithm for applying nonlinearities, closes a small, but important gap in the literature on equivariant networks for image and geometric data processing: It removes most architectural restictions, making the design of such networks significantly easier. The algorithm is easy to understand and implement; the only (but, as our experiments show, crucial) departure from baseline angular-domain evaluation is a resolution increase. Thereby, it achieves rotational stability sufficient for most practical applications at reasonable overhead. In critical applications, such as physical simulations, polynomial nonlinearities can be used to provide exact equivariance up to numerical precision (at the cost of minor losses in performance, in our experiments).

The biggest conceptual limitation of this approach is that it only applies to SO(2) and might be hard to generalize to more complex compact Lie groups like SO(3) (without using auxiliary orientation information), as harmonic distortion analysis in non-commutative symmetry groups becomes significantly more involved (requiring, for example, a matrix-valued convolution theorem).

## Acknowledgments and Disclosure of Funding

This work was supported by the Collaborative Research Center TRR 146 of the Deutsche Forschungsgemeinschaft (DFG). We thank Marc Stieffenhofer, Tristan Bereau and David Hartmann for fruitful discussions. We also would like to thank the anonymous reviewers for their many helpful comments and suggestions.

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
