# OpenReview forum: "General Nonlinearities in SO(2)-Equivariant CNNs"
_NeurIPS.cc/2021/Conference — NeurIPS 2021 Poster_

### Official Review · Reviewer_iJVc · 2021-07-12

**Rating:** 8
**Confidence:** 5

**Summary:**

The paper investigates the effect of nonlinearities in SO(2)-equivariant networks. When being applied in the spatial domain, such nonlinearities may lead to harmonic distortions, creating higher frequency harmonics in the signal. In a naive implementation this leads to aliasing effects, reducing the equivariance to a finite subgroup C_N < SO(2).
Due to their analytical tractability, the authors start by assuming nonlinearities that are polynomials of finite degree D. By employing the convolution theorem, they show that the Fourier coefficients of the nonlinearly transformed signal can be expressed in terms of convolutions of the input Fourier spectrum. This result implies in particular that inputs which are band limited to frequency K result in outputs that are band-limited to KD. Fully SO(2)-equivariant polynomial nonlinearities require therefore a D-fold oversampling of the input, to which the nonlinearity is applied pointwise.
For non-polynomial nonlinearities, the authors propose a D-fold oversampling, where D is a user-parameter. Equivariance is in this case approximate, but will decrease with increasing D.

The experimental section evaluates different nonlinearities and the effect of their hyperparameters.
The authors implement a G-CNN for G=SE(2), whose features can be seen as functions on fibers SO(2) over a base space R^2. As usual, the nonlinearities are acting on each fiber individually.
A second experiment implements SE(3)-equivariant Surfel networks, which are again relying on features which are functions on SO(2) at each point of a surface.
Figure 5 evaluates the equivariance of ReLU nonlinearities, which are non-polynomial. The relative equivariance error is hereby found to decay exponentially with the number of oversampled Fourier coefficients. The equivariance error of polynomial nonlinearities is in figure 6 shown to drop at the theoretically predicted cutoff to machine precision, which shows that they are indeed continuously SO(2)-equivariant. Classification results on the rotated MNIST dataset are approximately similar to those of previous work, however, the models require less parameters.
The surfel network achieves SOTA results on ModelNet-40, consuming a significantly reduced runtime in comparison to prior work.

**Ethical Concerns:**

There are no ethical issues with this submission.

**Limitations And Societal Impact:**

Limitations of the work are discussed appropriately. No immediate negative societal impacts are to be expected.


**Main Review:**

Originality:
SO(2)-equivariant CNNs in prior work relied on non-standard nonlinearities like tensor products and C-ReLUs or applied standard nonlinearities like ReLU to a discretization of the signal on a cyclic subgroup C_N. While the former perform usually unfavorably, the latter are not continuously SO(2)-equivariant. Inspired by harmonic distortion analysis, the authors propose polynomial nonlinearities which guarantee exact SO(2)-equivariance without compromising model performance or efficiency.
The method is new and related work is adequately cited.

Quality:
The submission is technically sound. All claims are well supported by the theoretical analysis and experimental results. Strengths and weaknesses are discussed appropriately.
Table 1 presents results in multiple different settings, however, the results of E2CNNs and the Poly networks are not immediately comparable since the number of numerical coefficients differ. It would be useful to add E2CNNs with 9 coefficients or Poly networks with 17 coefficients (or both).

Clarity:
The mathematical formulation is clear and the paper is well organized.
While the theoretical contribution is considering SO(2)-equivariant networks, the experimental section considers SE(2) and SE(3) equivariant models, which I found initially somewhat confusing. The authors should clarify how the models and the theory relate.
It would furthermore be helpful to clarify that the theory is developed from the viewpoint of regular SO(2)-CNNs with scalar features on SO(2) instead of steerable (representation valued) features. The authors should mention that their Fourier series representation of the features is equivalent to irrep features in steerable CNNs.
The linear maps W that are introduced in section 3 are not used in the experimental section, which may leave their role to some readers unclear. I could furthermore not follow the reasoning in lines 145-147 that w may additionally depend explicitly on a parameter beta.
Instead of the \otimes symbol in eqs. 9,10, I would suggest using \ast. This may help to avoid confusion since equivariant networks rely often on tensor products.
I do not understand the claim in lines 264,265 that the resulting angular functions may be interpreted as functions in the tangent plane. Should the angular functions not rather be identified with a function on unit-vectors?
The authors provide sufficient information to reproduce the empirical results.

Significance:
While the structure of linear equivariant maps is largely understood, the space of equivariant nonlinearities remained somewhat unclear. Specifically, the existing continuously SO(2)-equivariant nonlinearities Weiler and Cesa shown to perform inferior to discretized C_N-equivariant nonlinearities. The authors make an important contribution to the understanding of continuously SO(2)-equivariant nonlinearities.
The results are of particular importance for applications in natural sciences, where SO(2)-equivariance should in many cases hold exactly.


**Time Spent Reviewing:**

6

---

> ### Author Response · Authors · 2021-08-10
> **Response to Reviewer iJVc**
>
> We would like to thank Reviewer iJVc for the great amount of helpful feedback. In the following, we would like to address the questions and remarks brought up in the review.
>
> *Q: It would be useful to add E2CNNs with 9 coefficients or Poly networks with 17 coefficients (or both).*
>
> A: We agree that this would be useful to allow a better comparison. For the polynomial nonlinearities, we also measured the accuracy with 17 coefficients, however, the results (Poly2: 0.691%, Poly4: 0.694%) did not differ significantly from those obtained when using only 9 coefficients. The E2CNN paper does not provide results with a different number of coefficients for this specific architecture.
>
> *Q: The authors should clarify how the models and the theory relate.*
>
> A: The key difficulty is maintaining rotational equivariance, which is the theoretical contribution of our paper. The application in 2D and 3D networks described in the paper uses point-cloud networks (where rotational filters are co-moving with points), which is an established technique for obtaining perfect translational equivariance without much effort or complications. We should clarify this more prominently in the revision.
>
> *Q: It would furthermore be helpful to clarify that the theory is developed from the viewpoint of regular SO(2)-CNNs with scalar features on SO(2) instead of steerable (representation valued) features.*
>
> A: Yes, I think we should state this explicitly at the beginning of section 3.
>
> *Q: […] their Fourier series representation of the features is equivalent to irrep features in steerable CNNs*
>
> A: This is correct and should be explicitly mentioned for clarity. Viewing the representation as a set of Fourier coefficients is relevant for our way of applying non-linearities, as these do act simultaneously on a set of Fourier coefficients of different order, while most other approaches using irrep features apply non-linearities independently on each coefficient.
>
> *Q: The linear maps W that are introduced in section 3 are not used in the experimental section*
>
> A: Thanks for pointing this out, we will make more explicit that the matrices W are parametrized directly by the Fourier coefficients in our experiments.
>
> *Q: I could furthermore not follow the reasoning in lines 145-147 that w may additionally depend explicitly on a parameter beta*
>
> A: As all non-input layers operate on angle-dependent feature functions, the weights can depend on the angle of these functions (additional to the spatial angular dependency on the input image or point cloud). Though this dependency is not required to construct an equivariant network, it does increase performance, and is therefore also done in most other papers.
>
> *Q: I do not understand the claim in lines 264,265 that the resulting angular functions may be interpreted as functions in the tangent plane. Should the angular functions not rather be identified with a function on unit-vectors?*
>
> A: Correct, our formulation was too vague here. The angular functions can be interpreted as functions defined on tangential directions (i.e. unit-vectors tangential to the surface).

---

> > ### Comment · Reviewer_iJVc · 2021-08-17
> > **Reviewer response**
> >
> > Thank you for your detailed response. I will stick to my initial score of 8.

---

### Official Review · Reviewer_a9rF · 2021-07-15

**Rating:** 6
**Confidence:** 2

**Summary:**

The authors develop an SO(2)-equivariant convolution based on the Fourier transform. They show that non-linear activations create high frequencies and hence they clip those high frequencies in their convolution. They apply their method to both 2D data (rotated MNIST) and 3D data (rotated ModelNet40).

**Ethical Concerns:**

No.

**Limitations And Societal Impact:**

No mention.

**Main Review:**

The paper seems to have a very nice contribution, but it was quite hard for me to read it. The presentation of the method should be improved.

Table 1: Why is it important to have so many versions of the proposed method here? What is the best choice? Also - is there only one other paper that has results on rot-MNIST?

Please cite the paper "Vector Neurons: A General Framework for SO(3)-Equivariant Networks":
https://arxiv.org/pdf/2104.12229.pdf

Table 2: Some comparisons from papers cited in the paper above are missing, and there are some works there that out-perform the proposed method. See table 1 for ModelNet40.

Some typos:
line 145: "do" -> "to".
line 147: " show that this is not only the most case and but": please rephrase.
line 186: "anbular" -> angular
line 202: "by clip the" -> "by clipping the"

=========== Post rebuttal ==================

In light of the comments made by the other reviewers, I upgrade my score to 6. Regarding my comment on the missing comparisons to other methods: the author mention that their method is more stable, yet I do not see a discussion about that in the paper. How can you reach that your method is more stable than others? Is there a theoretical justification for it? Regarding your explanation: the better methods look pretty consistent wrt the two columns that you mention.

**Time Spent Reviewing:**

4

---

> ### Author Response · Authors · 2021-08-10
> **Response to Reviewer a9rF**
>
> We would like to thank Reviewer a9rF for the constructive feedback. On a general note, we also have the perception that the paper might be not easily accessible, in particular for readers not yet very familiar with the voluminous (and partially quite technical) prior work. It would thus be helpful to know in more detail which passages of our paper were difficult to understand, so we could improve these for our final revision. A few further hints would be highly appreciated.
>
> In the following, we will address the specific questions and remarks brought up in the review.
>
> *Q: Table 1: Why is it important to have so many versions of the proposed method here? What is the best choice?*
>
> A: As a main point of this paper was to evaluate the errors in equivariance of various nonlinearities when using the FFT-based algorithm, we felt that also providing the accuracy for the trained networks was important to show that our method actually works as intended (maintaining accurate equivariance would be not interesting if there were substantial performance losses; in the extreme case, a network outputting constant values would be perfectly equivariant and useless). Regarding the best choice of nonlinearity, we would argue that this depends on the application. If accurate equivariance is a requirement, then the polynomial nonlinearities are a good choice as they are accurate up to numerical precision by construction, while performance suffers only mildly. If approximate equivariance is sufficient, then the FFT-based algorithm allows to use any nonlinearity that could also be used in a non-equivariant network, which can be useful to reproduce existing neural network architectures in an equivariant scenario.
>
> *Q: Also - is there only one other paper that has results on rot-MNIST?*
>
> A: There are various other papers that perform evaluations on MNIST-rot, e.g. [5], [19], [33], [34] and [35]. To our knowledge, Weiler and Cesa [31] were the first to provide a comprehensive study on E(2)-Equivariant Steerable CNNs. To get meaningful results in our experiments on different nonlinearities, we chose to keep our model as close as possible to the best performing model from this study. As it was not our goal to compare different architectural properties (like number of layers, CNN vs ResNet, etc.), we did not feel that adding results achieved with different network architectures would add any value to this comparison.
>
> *Q: Please cite the paper “Vector Neurons: A General Framework for SO(3)-Equivariant Networks”*
>
> A: Yes, this reference would be a good addition, as it also investigates SE(3)-equivariant networks and places its main focus on how to handle nonlinearities (though their method differs from ours by using 3D vectors as features).
>
> *Q: Table 2: Some comparisons from papers cited in the paper above are missing, and there are some works there that out-perform the proposed method. See table 1 for ModelNet40.*
>
> A: Thanks for pointing this out. Some of the methods outperform our method in classification performance on ModelNet40, and they should be added to the comparison in Table 2 in our final revision. Still, it is important to emphasize that our method achieved more stable equivariance than the better performing methods cited in the named paper, as can be seen from their different accuracies in the SO(3)/SO(3) compared to the z/SO(3) benchmarks (which was our central objective).

---

> > ### Comment · Reviewer_a9rF · 2021-08-24
> > **Response**
> >
> > Thank you for your response. In light of the comments made by the other reviewers, I upgrade my score to 6.
> >
> > Regarding my comment on the missing comparisons to other methods: the author mention that their method is more stable, yet I do not see a discussion about that in the paper. How can you reach that your method is more stable than others? Is there a theoretical justification for it? Regarding your explanation: the better methods in the linked paper look pretty consistent wrt the two columns that you mention in the response.

---

> > > ### Author Response · Authors · 2021-08-30
> > > **Response**
> > >
> > > Thanks for your feedback. The notion of "method achieved more stable equivariance" could probably be easily misunderstood. We should thus discuss this more clearly: Our general goal is to maintain equivariance with an accurate up to (or close to) numerical precision. The methods in the "Vector Neurons"-Paper (Table 1 in that Reference) that perform a bit better than our method on ModelNet-40 (VN-DGCNN, SFCNN, LGR-Net) all show deviations between z-aligned (z/SO(3)) and unaligned (SO(3)/SO(3)) operation (in the range of 0.2%-5%). For some (in particilar, LGR-Net with 0.2%), the deviation is small and one would not expect a significant impact on classification tasks. However, there are applications where a deviation from equivariance (in this task specifically: invariance) of 0.2% is not good enough, e.g, when learning potential energies in phyiscal simulations that must remain invariant under changes of the reference frame.
> > >
> > > In our reply, we have referred to the accuracy at which equivariance is maintained as "stability" of equivariance, which might be misinterpreted. We will discuss this more carefully in the revised paper.

---

### Official Review · Reviewer_SC1j · 2021-07-16

**Rating:** 7
**Confidence:** 4

**Summary:**

The paper studies the effect of discretization issues when using different point-wise non-linearities (applied on the regular representation) in SO(2) equivariant neural networks. Unfortunately, the composition of band-limited SO(2) functions (in the features of an equivariant network) with popular point-wise non-linearities (such as ReLU, ELU or tanh) results in SO(2) functions whose spectrum is decaying but is not generally band-limited. A band-limited spectrum can be reconstructed up to desired precision using a larger number of samples but will always include some non-equivariant approximation error.

To guarantee perfect equivariance, the authors propose the use of polynomial activation functions, which guarantee band-limited output funtions and therefore enable perfect reconstruction of the spectrum using a finite number of samples as prescribed by the sampling theorem.

The authors compare different types of non-linearities on MNIST-rot, a classical benchmark for rotation equivariant models, and ModelNet40.
Their models achieve results comparable with the State-of-the-Art while achieving more stable equivariance.

**Limitations And Societal Impact:**

I described the main limitation in the MAIN REVIEW.

As the authors mentioned, this approach only applies to the SO(2) group and might be hard to immediately generalize to more general Lie groups due to their more complex harmonic analysis.
I only find this a minor limitation.

I don't see any issue with the authors' comments on the societal impact of their work.


**Main Review:**

I think the authors study a very relevant problem which affects many equivariant models in the literature. The work relies on standard tools from signal processing and harmonic analysis to clearly characterize this problem and to propose a solution. I also found the presentation clear overall.

I think the authors should also cite at least [1, 2] (see below), which also used SO(2) point-wise non-linearities based on Fourier Transform.

While I agree polynomial activations are necessary to achieve perfect equivariance, I find the relative error measured when using ELU or ReLU non-linearites sufficiently low for any practical application in a neural network. Indeed, Fig 6 shows that ELU achieves a *relative error* lower than 1% with ~12 samples. The final classification is anyways usually perfomed using an argmax over the output logits, which are usually trained using a cross entropy loss to favor a single class; I expect this to make the model sufficiently robust to small invariance errors in most cases. This also seems to agree with the MNIST-rot experiments in [31], where the performance of the C_N models saturate after N=12. I would like to see some more comments about this in the paper.




[1] RotDCF: Decomposition of Convolutional Filters for Rotation-Equivariant Deep Networks, Cheng et al.
[2] Gauge Equivariant Mesh CNNs Anisotropic convolutions on geometric graphs, de Haan et al.

Edit after rebuttal:
I have read the reviews and author responses, and decide to maintain the same rating.

**Time Spent Reviewing:**

2

---

> ### Author Response · Authors · 2021-08-10
> **Response to Reviewer SC1j**
>
> We would like to thank Reviewer SC1j for the constructive remarks and for pointing us to two more papers using Fourier transform for applying the non-linearities. We will include them in the revised version. For clarity we would like to add that while those papers apply non-linearities in a similar fashion than we do, they do not provide an analysis of how and when oversampling is able to preserve equivariance for various nonlinearities, which we consider our main contribution.
>
> Concerning the stability of the equivariance in practical applications, we concur that this point should be discussed in more detail in the paper. Specifically for classification tasks, we agree with the assessment that the approximate method should usually provide sufficiently low errors with common activation functions like ELU or ReLU. This is also supported by the measurements in Figure 5, which show an even lower error for the logits compared to the intermediate network layers. However, we think that for other applications, such as regression tasks, where no softmax cross-entropy-loss is applied to the output, lower error rates might be beneficial. Especially for applications in natural sciences, exact equivariance is an essential property.
>
> Finally, we fully agree that applying our approach the general Lie groups would certainly be much more involved.

---

> > ### Comment · Reviewer_SC1j · 2021-08-16
> > **Response**
> >
> > Thanks for the clarifications. I will maintain my rating as is

---

### Official Review · Reviewer_5pXb · 2021-07-16

**Rating:** 8
**Confidence:** 5

**Summary:**

The paper describes a focussed analysis of non-linearities to be used with steerable (Fourier based) implementations of equivariant neural networks. The authors show how to get exact equivariance with polynomial activation functions and provide guidelines on how to use general activation functions via the route [Inverse Fourier > activation func > Fourier]. Their theoretical insights are confirmed by experimentation.

**Ethical Concerns:**

I see none.

**Limitations And Societal Impact:**

I think limitations with respect to generalization of the result to more realistic datasets could have been addressed (see MNIST critic above). Otherwise, I think the scope is clearly given and appropriate.

**Main Review:**

The paper is very well written (up to several typos, see minor comments below) and has a high tutorial value. The paper also comes with code, which further increases its usefulness.

While most of the notions addressed in this paper (equivariance and numerical artifacts due to band-limiting or discretization) have been addressed in several other works, it is very valuable to have the essentials in one place and clearly presented. The citations to other works seem accurate and appropriate.

The experiments are sound, though perhaps somewhat limited to easy problems such as MNIST. I can very well imagine that conclusions drawn on MNIST do not generalize to other datasets because the the dataset is so simple. This is my main concern, and it could be addressed by validating on a more representative dataset (STL10, tiny image net, ...) in which equivariance is I think also more important.

Finally, after reading the paper I am missing a short discussion on the relevance of approximate vs exact equivariance in the context of performance. Namely, ReLU based methods still outperform the exact activation functions. What does this mean? Apparently it is OK to be not perfectly equivariant, but does this mean ReLU is still the best activation function, or does it mean equivariance errors can be exploited to get better performance? Since the main focus of the paper is on achieving exact equivariance through appropriate activation functions I feel a short discussion regarding the above is in place.

All in all, I think this is a valuable paper and only have some small remarks left:

I got a bit confused after equation 5. The idea of output coordinate dependency of the convolution kernel seems to have been analyzed before (ref 31), however, I don't get it. It seems like a very unusual thing to do. Do I understand correctly that usually one does not have beta dependency and thus w_{i,\beta}=w_{i,0} for every beta? Are there other works actually using this. Is this paper using it?

Related is the interpretation of equation 6. It is introduced as a point-wise multiplication, but then in the equation I see a sum over a frequency index which to me looks more like an inner product. Does this have to do anything with the beta dependency of the conv kernel? In a point-wise multiplication I wouldn't expect a reduction over the points... Could this please be clarified.

Also, I don't see how the beta dependency is used in the paper. Could it be mentioned whether this is the case or not? I think the reader (me at least) would be helped with emphasizing the simpler case of no beta dependency.

Line 195 "we can also expect favorable numerical properties". Favorable compared to what?

Some typos:
line 55 "compute the a"
line 145 "correspond do convolution"
line 147 "most case and but also"
line 156 "is still holds"
line 160 "non-linear can"
line 172 "outputs will"
line 186 "anbular"
line 201 "problem of is"
line 202 "by clip the"
and probably several more... please check this.


**Time Spent Reviewing:**

8

---

> ### Author Response · Authors · 2021-08-10
> **Response to Reviewer 5pXb**
>
> We would like to thank Reviewer 5pXb for the constructive and helpful feedback. We agree that MNIST is a rather simple data set and the impact of equivariance on classification accuracy might be more visible for other data sets. The goal of our paper is not to maximize classification accuracy but to understand how to obtain networks with accurate equivariance (which is more relevant for applications beyond image classification, such as predicting potentials and forces in computational physics). Still, we opted to perform the 2D experiments on MNIST-rot, as it is the most widely used dataset for experiments on SO(2)-equivariance and allows us to compare our results (in terms of accuracy of equivariance) to other papers. We hoped to mitigate the concerns on generalizability by choosing ModelNet40 as an additional dataset for our experiments, which is more challenging than MNIST and also widely used in papers covering SO(3)-equivariance.
>
> In the following, we would like to address the questions and issues brought up in the review.
>
> *Q: I am missing a short discussion on the relevance of approximate vs exact equivariance in the context of performance.*
>
> A: We agree that this should be discussed in more detail, and we will add a discussion to the revised version. As ReLU generally works well as nonlinearity in neural networks, it is no surprise that it also works well in our equivariant setting in cases where approximate equivariance is good enough for typical classification problems. For the classification tasks on MNIST and ModelNet40, the approximate FFT method provides stable enough equivariance, as a larger oversampling, while it does indeed reduce the mathematical error (see Figure 6), has negligible impact on classification accuracy (see Table 1). As mentioned above, accurate equivariance becomes more important for other applications such as approximate physics simulation – we use classification benchmarks in our paper because this permits a broad comparison (while it is no problem to measure the precision of equivariance by comparing intermediate layer outputs).
>
> *Q: […] does it mean equivariance errors can be exploited to get better performance?*
>
> A: Exploiting equivariance errors is a valid concern if the input data is not augmented by random rotations, thereby relying only on the equivariance properties of the network itself. In fact, architectures that are not exactly equivariant often show better performance in this scenario. To avoid such exploitation of equivariance errors, we always applied random rotations to the input examples before passing them to the network when measuring classification accuracy.
>
> *Q: I got a bit confused after equation 5. […] Do I understand correctly that usually one does not have beta dependency and thus w_{i,\beta}=w_{i,0} for every beta? Are there other works actually using this. Is this paper using it?*
>
> A: We agree that our formulation here is difficult to understand and will try to clarify this in the revised paper. Equation 5 describes a common optimization from literature to the linear part of an equivariant layer. While the input layer of an equivariant network usually operates on scalar features (as common datasets usually contain scalar values), the weights for this layer only depend on one angle (the angular filter response in the spatial domain of the input). However, all following layers operate on feature functions instead of scalars that are also angle-dependent. In this case, the weights can additionally depend on the angle of those feature functions. This essentially allows the network to “see” the response to all rotations in the previous layers, which is required to take full advantage of the equivariant architecture. Though this additional angular dependency is not required for equivariance alone, it is required to reach comparable performance, as most other papers also use this approach. Importantly, the dependency input angle does not affect the behavior of the nonlinearities (the central topic of our paper), which operate on functions dependent on the *output* angle only.
>
> *Q: Related is the interpretation of equation 6. It is introduced as a point-wise multiplication, but then in the equation I see a sum over a frequency index which to me looks more like an inner product. Does this have to do anything with the beta dependency of the conv kernel?*
>
>
> A: This is related/adding to the discussion above. To clarify: Yes, this is because W depends on two angles. As a consequence, the Fourier transform of W is a 2D Fourier transform, where the coefficients can be represented as a 2D matrix, here indexed by k and k’. Summation is done on one of the axes, to eliminate the dependencies on two angles and yield an output that only depends on one angle.
>
> *Q: Also, I don't see how the beta dependency is used in the paper. Could it be mentioned whether this is the case or not?*
>
> A: We use the full size weight matrix depending on the two angles in our experiments (expect for the first layer, where this is not applicable as it has scalar inputs).
>
> *Q: Line 195 "we can also expect favorable numerical properties". Favorable compared to what?*
>
> A: This should probably be reworded: Favorable was meant here with respect to the Fourier transform being a unitary map, which avoids numerical instabilities that other (non-unitary) transformations might produce.

---

> > ### Comment · Reviewer_5pXb · 2021-08-20
> > **Response**
> >
> > Thank you for the clarifications. I especially appreciate the explanation on the output angle dependency of the layers. I think it is worthwhile to put some extra intuition in text. Also I believe either a cite to an explicit location would help. E.g. I couldn't quite found out where in the Weiler-Cesa paper to look. *A proof in the suppl. may be even better.*
> >
> > I namely still do not get the intuition of why this is possible. E.g. if I substitute $\alpha \rightarrow \alpha - \gamma$ as a rotation on the input, I get that the map is equivariant
> >
> > $(K(\mathcal{L}_\gamma x)(\beta) = K(x)(\beta - \gamma)$
> >
> > iff $w$ does not depend on $\beta$ (because in the r.h.s it would have to depend on $\beta - \gamma$), here ${L}_{\alpha} x(\alpha) = x(\alpha - \gamma)$ .
> >
> > If there is somehow a dependency on $\beta$ then it means that the map is no-longer linear but perhaps more something like an attentive group convolution in which there is an input (or output) dependency via an attention map that depends on the input in an equivariant manner. See e.g. [Romero et al. 2020]. Is this how I should look at it? I can imagine that equation 6 implements something like an attentive group convolution in the Fourier domain, with the attention function being a linear transformation of the input.
> >
> > [Romero et al. 2020] David Romero, Erik Bekkers, Jakub Tomczak, Mark Hoogendoorn Proceedings of the 37th International Conference on Machine Learning, PMLR 119:8188-8199, 2020.
> >
> >
> > So, even though it is not a main concern for the paper, the output dependency topic is still a bit vague to me and an intuitive explanaition or proof would be appreciated. I still think highly of the paper and do not intent to change my score.

---

> > > ### Author Response · Authors · 2021-08-30
> > > **Response**
> > >
> > > Thanks for the helpful feedback. In the following, we provide a more detailed derivation of our method for the SO(2)-equivariant case. We agree with the idea to add this derivation to the supplementary material, and hope that it gives the readers a better understanding of how our method works and answers your questions on the angle-dependency of the weights.
> > >
> > > As noted before, the activations in our equivariant layers are angle-dependent functions. In the continuous case, this would mean that an activation map $x$ simultaneously depends on a location $\textbf{u} \in \mathbb{R}^2$ and an angle $\alpha \in \textrm{SO(2)}$. To achieve equivariance, a joint convolution over these two domains is required:
> > >
> > > $
> > > x^{(l)}\left(u,\alpha\right)=\varphi\left(\int_{\mathbb{R}^2} \int_0^{2\pi} W^{(l)}\left(\Theta_\alpha (\textbf{u}'-\textbf{u}), \alpha'-\alpha\right)  x^{(l-1)}\left(u', \alpha'\right) d\alpha' d\textbf{u}'\right)
> > > $
> > >
> > > This is explained in detail in the paper "RotDCF: Decomposition of Convolutional Filters for Rotation-Equivariant Deep Networks" (see equation 2 in that paper). We use $\varphi$ to represent the nonlinearity. It can be verified that this approach provides equivariance by replacing the input activations $x^{(l-1)}(\textbf{u}',\alpha')$ by $x^{(l-1)}(\Theta_\gamma \textbf{u}',\alpha'-\gamma)$, yielding the same transformation for the output. For simplicity, we left out channel index and the summation over input channels in the above equation (i.e. we consider the case of only one input and one output channel).
> > >
> > > As we work on point clouds, we consider activations $x_1^{(l)}$, ..., $x_N^{(l)}$ to be Dirac peaks located at specific points $\textbf{p}_1^{(l)}$, ..., $\textbf{p}_N^{(l)}$. The integration over $\mathbb{R}^2$ thus becomes a sum over the points in the previous layer:
> > >
> > > $
> > > x_j^{(l)}\left(\alpha\right)=\varphi\left(\sum_{n=1}^{N^{(l-1)}} \int_0^{2\pi} W^{(l)}\left(\Theta_\alpha\left(\textbf{p}_n^{(l-1)}-\textbf{p}_j^{(l)}\right), \alpha'-\alpha\right) x_n^{(l-1)}\left(\alpha'\right) d\alpha'\right)
> > > $
> > >
> > > To be able to use Fourier representations, we decompose our trainable weights $W$ into a product of an angular part $Q$ and a radial part $R$:
> > >
> > > $
> > > x_j^{(l)}\left(\alpha\right)=\varphi\left(\sum_{n=1}^{N^{(l-1)}} \int_0^{2\pi} R^{(l)}\left(d_{n,j}^{(l)}\right) Q^{(l)}\left(\beta_{n,j}^{(l)}+\alpha, \alpha'-\alpha\right) x_n^{(l-1)}\left(\alpha'\right) d\alpha'\right)
> > > $
> > >
> > > with the distance $d_{n,j}^{(l)}=\lVert \textbf{p}_n^{(l-1)}-\textbf{p}_j^{(l)} \rVert$
> > >
> > > and the relative angle $\beta_{n,j}^{(l)}=\angle(\textbf{p}_n^{(l-1)}-\textbf{p}_j^{(l)})$.
> > >
> > > The activations are represented in a Fourier basis:
> > >
> > > $
> > > x_j^{(l)}\left(\alpha\right)=\sum_k z_{j,k}^{(l)} e^{ik\alpha}
> > > $
> > >
> > > Similarly, the angular part of the weights is represented in a 2D Fourier basis (as it depends on two angles):
> > >
> > > $
> > > Q^{(l)}\left(\beta, \alpha\right)=\sum_u \sum_v q_{u,v}^{(l)} e^{iu\alpha} e^{iv\beta}
> > > $
> > >
> > > Plugging in the Fourier representations on the right hand side gives:
> > >
> > > $
> > > x_j^{(l)}\left(\alpha\right)=\varphi\left(\sum_{n=1}^{N^{(l-1)}} \sum_u \sum_v \sum_k \int_0^{2\pi} R^{(l)}\left(d_{n,j}^{(l)}\right) q_{u,v}^{(l)} e^{iu(\alpha'-\alpha)} e^{iv(\beta_{n,j}^{(l)}+\alpha)}  z_{n,k}^{(l-1)} e^{ik\alpha'} d\alpha'\right)
> > > $
> > >
> > > Which can be reordered to:
> > >
> > > $
> > > x_j^{(l)}\left(\alpha\right)=\varphi\left(\sum_{n=1}^{N^{(l-1)}} R^{(l)}\left(d_{n,j}^{(l)}\right) \sum_u \sum_v \sum_k q_{u,v}^{(l)} e^{i(v-u)\alpha} e^{iv\beta_{n,j}^{(l)}} z_{n,k}^{(l-1)} \int_0^{2\pi} e^{i(u+k)\alpha'} d\alpha'\right)
> > > $
> > >
> > > As $\int_0^{2\pi} e^{i(u+k)\alpha'} d\alpha' = \delta_{u,-k}$ we can drop the sum over $u$ and replace $u$ with $-k$:
> > >
> > > $
> > > x_j^{(l)}\left(\alpha\right)=\varphi\left(\sum_{n=1}^{N^{(l-1)}} R^{(l)}\left(d_{n,j}^{(l)}\right) \sum_k \sum_v q_{-k,v}^{(l)} z_{n,k}^{(l-1)} e^{i(v+k)\alpha} e^{iv\beta_{n,j}^{(l)}}\right)
> > > $
> > >
> > > Replacing $v$ with $v-k$ (sum shift) gives:
> > >
> > > $
> > > x_j^{(l)}\left(\alpha\right)=\varphi\left(\sum_{n=1}^{N^{(l-1)}} R^{(l)}\left(d_{n,j}^{(l)}\right) \sum_k \sum_v q_{-k,v-k}^{(l)} z_{n,k}^{(l-1)} e^{iv\alpha} e^{i(v-k)\beta_{n,j}^{(l)}}\right)
> > > $
> > >
> > > This gives $e^{iv\alpha}$ on the right hand side, which allows us to extract the output Fourier coefficients:
> > >
> > > $
> > > z_{j,v}^{*(l)}\left(\alpha\right)= \sum_{n=1}^{N^{(l-1)}} R^{(l)}\left(d_{n,j}^{(l)}\right) \sum_k q_{-k,v-k}^{(l)} z_{n,k}^{(l-1)} e^{i(v-k)\beta_{n,j}^{(l)}}
> > > $
> > >
> > > Note that we use $z^*$ instead of $z$ here to denote the output Fourier coefficients before application of the nonlinearity $\varphi$ (in our case, the nonlinearity always works on a set of Fourier coefficients simultaneously and cannot be represented as a function being applied to a single coefficient).
> > >
> > > The equation above shows that an output Fourier coefficient of a specific rotation order can depend on all input rotation orders (sum over v), where in each case a correct spatial filter must be used to preserve equivariance (i.e. the angular part of the spatial filter is fixed to $e^{i(v-k)\beta_{n,j}^{(l)}}$). The angular-dependent part $q$ of the weights depends on both the input and output rotation order. Our approach of applying the convolution is not attention-based, but equivalent to the irrep approach used in the paper by Weiler and Cesa, as correctly pointed out by reviewer iJVc. However, our method differs in the way the nonliearities are applied.

---

> > > > ### Comment · Reviewer_5pXb · 2021-08-31
> > > > **Thanks!**
> > > >
> > > > Thanks this has been very helpful! Adding it as suppl. mat. would be greatly appreciated.
> > > >
> > > > I understand this derivation, though it was not clear from the paper. Especially as the intuition is now obtained from regular SE(2) group convolutions towards Fourier based (steerable) g-convs. Namely, section 3 only talks about functions on SO(2) and not $R^2 \rtimes SO(2)$ as in the discussion above and the $\beta$ is referred to as output angle, whereas in the discussion above it relates to the angular part of the spatial part of the kernel.
> > > >
> > > > I suppose the point is that such a mapping may more generally depend on an interaction between two angular functions (in the example above a spatial part and an angular part) to produce a new SO(2) signal. I have the impression that this follows the same structure of the SE(3) steerable methods that rely on the Clebsch-Gordan product for mixing of steerable vectors (Fourier coefficients) and that this is a 2D variation of it. Please excuse my ignorance, I found this connection intriguing but can't quite put my finger on it.
> > > >
> > > > I don't mean to keep this discussion going on forever and am already very content with your current answers. Don't feel obliged to respond.

---

> > > > > ### Author Response · Authors · 2021-09-02
> > > > > **Thanks**
> > > > >
> > > > > We agree that section 3 should be updated to make the simultaneous dependency of the convolution on $R^2$ and SO(2) clear at this point, in addition to the derivation in our previous response which we plan to add to the supplementary material of the paper. Regarding the similarity of our approach to SE(3) based steerable methods based on Clebsch-Gordan products: we find this quite interesting, but we are also not sure how exactly they are connected.
> > > > >
> > > > > Finally, thanks again for your helpful and constructive feedback!

---

### Decision · Program_Chairs · 2021-09-27

**Decision:**

Accept (Poster)

**Comment:**

Congratulations, the paper is accepted to NeurIPS 2021!
Please add a discussion on tradeoffs of approximate versus exact equivariance.
Furthermore, please add additions to the supplementary as discussed with reviewer 5pXb.
Please consider to include/elaborate the discussion on the challenges in generalizing your method to other transformation groups.
Lastly, please incorporate other suggestions and edits as discussed in rebuttal and reviews.